# Tropical volcanism enhanced the East Asian summer monsoon during the last millennium

Fei Liu [1], Chaochao Gao [2✉], Jing Chai[3,4], Alan Robock [5], Bin Wang [6,7✉], Jinbao Li [8], Xu Zhang [9], Gang Huang [4] & Wenjie Dong[1]

Extreme East Asian summer monsoon (EASM) rainfall frequently induces floods that threaten millions of people, and has been generally attributed to internal climate variability. In contrast to the hydrological weakening theory of volcanic eruptions, here we present convergent empirical and modeling evidence for significant intensification of EASM rainfall in response to strong tropical volcanic eruptions. Our multi-proxy analyses show a significantly increased EASM in the first summer after tropical eruptions from 1470 AD to the present, and the more frequent occurrence of El Niños in the first boreal winter after eruptions is necessary for the enhanced EASM. Model simulation ensembles show that a volcano-induced El Niño and the associated stronger than non-volcanic El Niño warm pool air-sea interaction intensify EASM precipitation, overwhelming volcanic-induced moisture deficiency. This work sheds light on the intertwined relationship between external forcing and internal climate variability and potential flood disasters resulting from tropical volcanic eruptions.

[1] School of Atmospheric Sciences Sun Yat-Sen University, Key Laboratory of Tropical Atmosphere-Ocean System Ministry of Education, and Southern Marine Science and Engineering Guangdong Laboratory, Zhuhai 519082, China. [2] College of Environmental and Resource Sciences, Zhejiang University, Hangzhou 310058, China. [3] Plateau Atmosphere and Environment Key Laboratory of Sichuan Province, School of Atmospheric Sciences, Chengdu University of Information Technology, Chengdu 610225, China. [4] State Key Laboratory of Numerical Modeling for Atmospheric Sciences and Geophysical Fluid Dynamics, Institute of Atmospheric Physics, Chinese Academy of Sciences, Beijing 100029, China. [5] Department of Environmental Sciences, Rutgers University, New Brunswick, NJ 08901, USA. [6] Department of Atmospheric Sciences and International Pacific Research Center, University of Hawaii at Manoa, Honolulu, HI 96822, USA. [7] Earth System Modeling Center and Climate Dynamics Research Center, Nanjing University of Information Science & Technology, Nanjing 210044, China. [8] Department of Geography, University of Hong Kong, Hong Kong SAR, China. [9] Alpine Paleoecology and Human Adaptation Group (ALPHA), State Key Laboratory of Tibetan Plateau Earth System, Resources and Environment (TPESRE), Institute of Tibetan Plateau Research, Chinese Academy of Sciences, Beijing, China. ✉email: gaocc@zju.edu.cn; wangbin@hawaii.edu

The record-breaking EASM rainfall in 2020 caused a humanitarian disaster, affecting 45.4 million people in China and resulting in over 142 human casualties and more than 16 billion US dollars in economic loss[1]. EASM rainfall is usually enhanced by moisture advection and convergence due to southwesterly anomaly of an enhanced western North Pacific subtropical high, following an El Niño peak in the previous winter[2–6]. A recent study also proposed that El-Niño-related tropical tropospheric warming could shift the midlatitude westerlies southward, which would impinge on the Tibetan Plateau and induce northerlies downstream of the plateau to intensify the East Asian early summer rainband[7].

Large volcanic eruptions are one of the major natural external forcings of Earth's climate variability from interannual to centennial time scales[8,9]. The EASM is found to be suppressed after volcanic eruptions in model simulations[10–12] due to the reduction of surface shortwave radiation and the associated slowdown of the global hydrological cycle[13–16]. Volcanic eruptions have been suggested as the cause of some historical drought events[17–19]. However, no significant drought was recorded in the Chinese chronicles following the large 1815 Tambora eruption[20], nor in observations following the 1982 El Chichón and 1991 Pinatubo eruptions[21]. These case studies raise a question about the perception of EASM suppression by volcanic perturbations.

Previous studies of the effects of volcanic eruptions on EASM commonly overlook the role of El Niño, although it has been suggested by a number of proxy-reconstructed El Niño/Southern Oscillation (ENSO) indices that the likelihood of an El Niño increases after tropical eruptions[22–28]. Two main mechanisms have been proposed using state-of-the-art model simulations to explain this connection: the ocean dynamic thermostat mechanism[29–31] and the land-sea thermal contrast mechanism[32–34].

This work investigates the response of the EASM-tropical Pacific system to tropical volcanism to understand the interaction and relative roles of internal climate feedback and external forcing. We analyze long-term multi-proxy data and multi-model simulations and find that a volcano-induced El Niño and the associated warm pool air-sea interaction can intensify the EASM precipitation, overshadowing the hydrological weakening by volcanic eruptions.

## Results

**Proxy evidence of enhanced EASM rainfall following eruptions**. We first examine the EASM-tropical Pacific response to 22 precisely dated tropical large volcanic eruptions[35] for the period from 1470 to 1999 AD using the gridded precipitation proxy data[36] and the ensemble mean of 11 sets of ENSO paleoclimate reconstructions (see "Methods").

The composite results of East Asian rainfall reconstruction in the first post-eruption summer show significant (at the 90% confidence level) positive precipitation anomalies centered around the Yangtze River basin, northeast China, and Kyushu of Japan, accompanied by negative, albeit not significant, anomalies over the Indo-China Peninsula and the Philippines (Fig. 1a). This meridional dipole precipitation anomaly pattern resembles the enhanced EASM-Pacific High response in the first post-El Niño summer, which is well observed with instrumental observations[2,4,5]. Enhanced EASM precipitation in the first post-eruption summer is also observed in two other well-known precipitation reconstructions over China[37] and over all of Asia[38] (Supplementary Fig. 1). Significant positive EASM (25°–34°N, 106°–122°E) precipitation anomalies (at the 95% confidence level), mainly over the middle-lower reach of Yangtze River, are only observed in the first boreal summer after the eruption (Fig. 1b).

**Strengthened El Niño-EASM relationship**. The ensemble mean of 11 available ENSO reconstructions indicates a significant post-eruption precipitation increase following an El Niño (Fig. 2a), in contrast to the normal EASM precipitation anomalies after an eruption without a preceding El Niño (Fig. 2b). Significantly negative anomalies, on the other hand, are observed over the Indo-China Peninsula, Taiwan, and the Philippines, demonstrating compelling evidence for an El Niño-enhanced Pacific High and its role in modulating the volcanic-induced EASM response. For the non-volcanic El Niño (Fig. 2c), the EASM precipitation increase is much weaker than that after a volcano-induced El Niño. Our multi-proxy analysis results thus demonstrate that explosive tropical eruptions tend to strengthen the El Niño-EASM relationship and increase EASM rainfall in the first subsequent boreal summer, suggesting seasonal monsoon-ocean feedback after large tropical eruptions.

**Multi-model evidence of enhanced EASM following eruptions**. Next, we investigate the last-millennium climate simulation results of 13 different models from the Paleoclimate Modeling Intercomparison Project phases 3 and 4 (PMIP3 & PMIP4) to interpret the monsoon-ocean response to tropical large volcanic eruptions in the context of the current state-of-the-art models. Based on the actual volcanic forcing used in each model, the 13 simulations amount to 92 eruption events for the common pre-industrial period from 1470 to 1849 AD, and they are used to construct a superposed epoch analysis (see "Methods"). The average of these 92 eruption simulations shows a relative El Niño signal, i.e., a reduced zonal gradient of Pacific equatorial sea surface temperature (SST), in the first post-eruption boreal winter (Supplementary Fig. 2a). In the first post-eruption summer, Indo-Pacific oceanic cooling and Asian drying are simulated, and maximum drying is found over the western Indo-China peninsula (Fig. 3a). Positive precipitation anomalies are simulated over the middle-lower reach of Yangtze River but are significant only in a small area. The associated EASM circulation index, defined by the lower-tropospheric zonal wind shear vorticity of the enhanced Pacific High (see "Methods"), exhibits a significantly negative anomaly, demonstrating an enhanced EASM circulation in the multi-model ensemble mean results (Fig. 3a).

The responses of EASM precipitation to the direct volcanic forcing and to an El Niño can be differentiated by comparing two sets of composites with or without the preceding El Niño among the 92 events (Supplementary Fig. 2). In the first post-eruption summer (Fig. 3b), the composite for 65 out of the 92 events without a preceding El Niño mainly exhibits a direct cooling and drying volcanic effect over most parts of the globe. A relative El Niño signal is simulated and the reduced zonal SST gradient across the Pacific weakly enhances the Pacific High, represented by an anticyclonic anomaly mainly over the South China Sea. Studies have established that SST gradients across the tropical Pacific strongly influence global rainfall[39]. Thus, an El Niño-like response in non-El Niño events tends to offset the cooling-induced dry monsoon over East Asia, resulting in an insignificant decrease in the EASM rainfall, consistent with reconstructions (Fig. 2b). In the presence of a preceding El Niño, the model results show increased post-eruption summer precipitation in the EASM region, and the most significant enhancement is located over the middle-lower reach of the Yangtze River, driven by the convergence and moisture advection of southwesterly wind anomalies of the enhanced Pacific High and extratropical northerly anomalies, albeit not significant, downstream of the Tibetan Plateau (Fig. 3c). Thus, the post-eruption El Niño cases can overwhelm the no-post-eruption El Niño cases, resulting in an overall increase in the EASM rainfall.

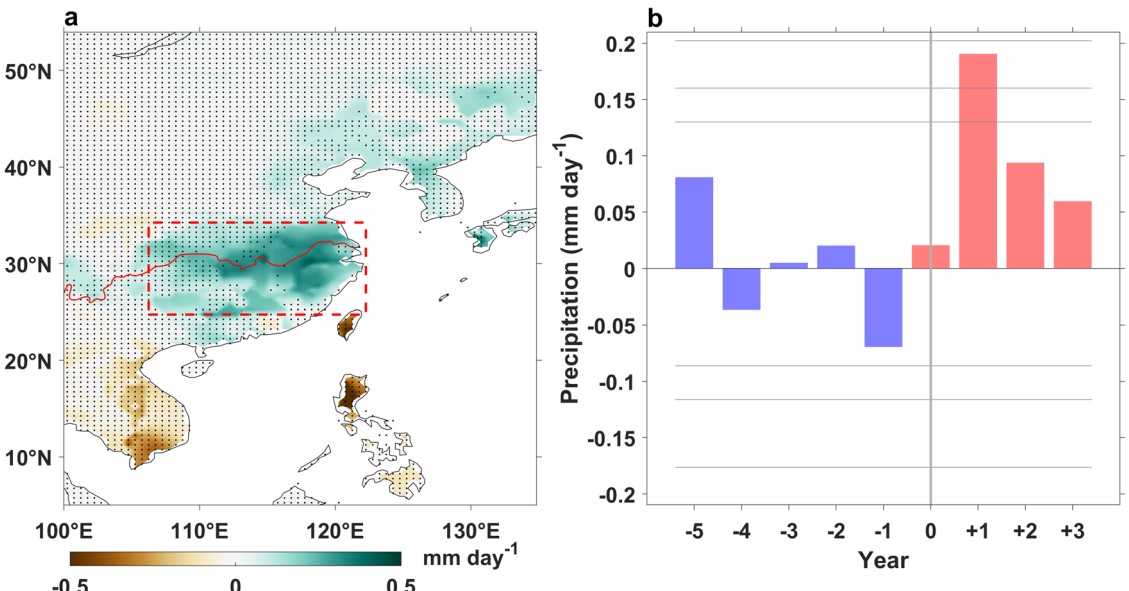

**Fig. 1 East Asian summer monsoon (EASM) response to tropical eruptions based on reconstructions. a** Superposed epoch analysis results of East Asian precipitation anomalies (shading) in the first boreal summer after 22 tropical eruptions from 1470 to 1999 AD. Stippling indicates precipitation anomalies not significant at the 90% confidence level. The red curve is the Yangtze River, and the red box indicates the EASM region (25°–34°N, 106°–122°E). **b** Composite EASM-averaged precipitation anomaly for 22 tropical eruptions. Confidence limits (90, 95, and 99%) are marked by horizontal lines. Red and blue colors mark the post-eruption and pre-eruption composites, respectively. Year 0 denotes the eruption year. This figure was created using MATLAB 2020a (http://www.mathworks.com).

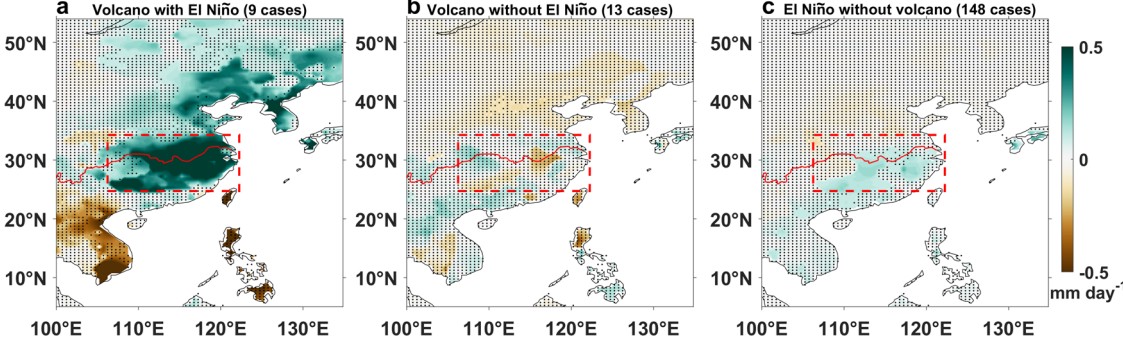

**Fig. 2 Role of El Niño in post-eruption EASM change based on reconstructions.** Superposed epoch analysis results of East Asian precipitation anomalies (shading) in the first boreal summer after **a** 9 tropical eruptions with and **b** 13 eruptions without an El Niño in the first winter after the eruption, and after **c** 148 non-volcanic El Niños during the period of 1470–1999 AD. An El Niño event is defined by the average of the 11 reconstructions of the ENSO index (see "Methods"). Stippling indicates precipitation anomalies not significant at the 90% confidence level. The red curve is the Yangtze River, and the red box indicates the EASM region. This figure was created using MATLAB 2020a (http://www.mathworks.com).

This El Niño-increased EASM precipitation after volcanic eruptions is also simulated when considering all 228 eruption events during the last millennium covering 850–1849 AD in all 13 PMIP models (Supplementary Fig. 3). Ten Community Earth System Model (CESM) last-millennium ensemble simulations from 1470 to 1999 AD also confirm the critical role of El Niño after volcanic eruptions in increasing EASM precipitation (Supplementary Fig. 4).

**Volcanic-induced Pacific air-sea interaction**. Eruption-induced radiative cooling leads to an El Niño in the post-eruption winter or an El Niño-like response in the following summer when the El Niño is not fully developed. The responses are determined by the coupled atmosphere-land-ocean dynamics. The decrease in SST gradient, i.e., relative warmer eastern Pacific, can be initiated by eruption-induced global cooling through the ocean dynamic thermostat mechanism[29,40] and land-sea thermal contrast[32,33].

This decreased SST gradient gives rise to a weakened pressure gradient and hence weaker easterly winds and Walker circulation (Supplementary Fig. 2a), which may, in turn, reduce the SST gradient, a mechanism known as "the Bjerknes feedback"[33,41].

Among the 92 eruption events, the post-eruption EASM precipitation change is significantly correlated with the preceding El Niño (0.19, at the 90% confidence level) two seasons before, and the correlation is higher for the EASM circulation (−0.44, at the 99% confidence level) than for the precipitation. This is because the EASM circulation index, i.e., the Pacific High, is better captured by numerical models than precipitation[42]. These significant relations indicate that a stronger El Niño tends to induce a larger EASM increase, consistent with the instrumental observations[43].

After a volcano-induced El Niño, strong negative precipitation anomalies are simulated over South Asia, consistent with previous studies demonstrating the drying effect of large volcanic eruptions on the South Asian monsoon[44,45]. The enhanced Pacific High is

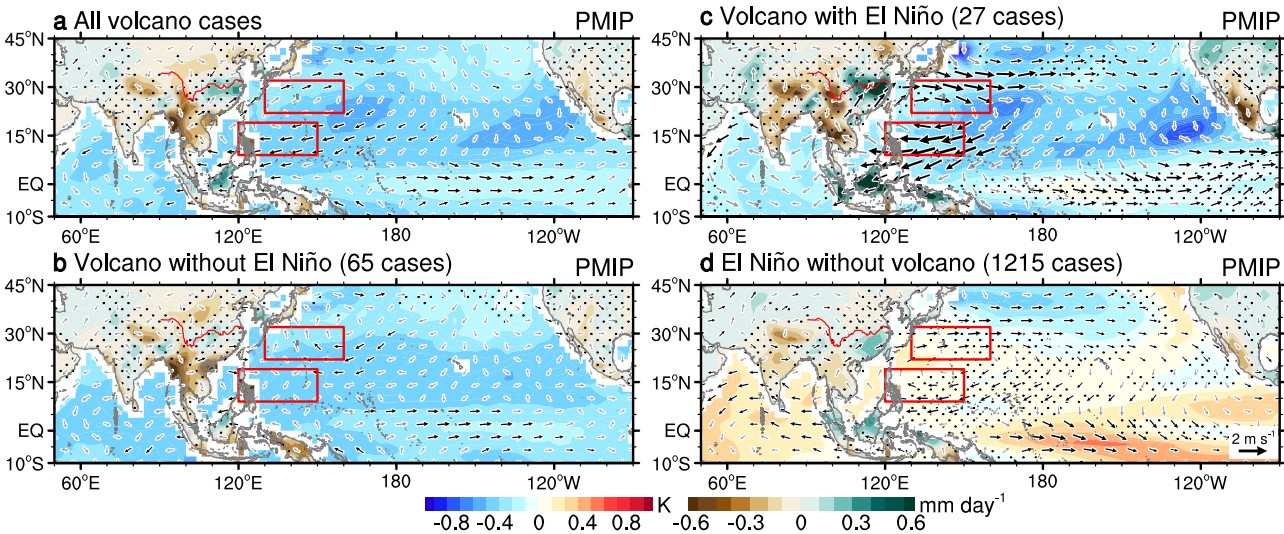

**Fig. 3 Simulated EASM-ocean responses to tropical eruptions.** Composite SST anomalies (shading over ocean), precipitation anomalies (shading over land), and 850 hPa wind anomalies (vectors) in the first boreal summer after **a** all 92 simulated tropical eruptions, **b** 65 eruptions without, and **c** 27 eruptions with an El Niño response in the first boreal winter after the eruption, as well as **d** composites for 1215 non-volcanic El Niños in 13 PMIP last millennium simulations from 1470 to 1849 AD. Stippling and gray vectors indicate precipitation and temperature anomalies and wind anomalies not significant at the 90% confidence level, respectively. The red curve is the Yangtze River. The red rectangles denote the locations where the EASM circulation index is defined: the 850 hPa zonal wind averaged in the southern box minus that in the northern box (see "Methods"). A negative value of this index represents enhanced Pacific high and associated East Asian subtropical rainfall. Maps created with The NCAR Command Language (Version 6.6.2) [Software]. (2019). Boulder, Colorado: UCAR/NCAR/CISL/TDD. https://doi.org/10.5065/D6WD3XH5.

tied to the SST cooling to its southeast (Fig. 3c), explained by the wind-evaporation SST feedback after an El Niño[4]. Easterly wind anomalies over the North Indian Ocean, which were also suggested to increase the Pacific High[5], are only significant over the Arabian Sea. Volcanic eruptions act to enhance this post-El Niño western North Pacific SST cooling and intensify the Pacific High (Fig. 3d), as found by previous simulations[26,46]. The resulted EASM enhancement is much larger than a non-volcanic El Niño would induce, confirming the proxy-finding of volcanic strengthened El Niño-EASM relationship.

The change of monsoon precipitation can be divided into dynamic and thermodynamic causes[47], which are related to the changes in circulation and moisture availability, respectively (see "Methods"). After a volcanic eruption, dynamic-related precipitation is increased while the thermodynamic-related precipitation is decreased, demonstrating the volcanic-induced cooling and drying (Fig. 4). The decreased thermodynamic-induced precipitation counteracts the increased dynamic-induced precipitation, resulting in a weak change in total precipitation. The presence of an El Niño tends to enhance the dynamic-related precipitation, while keeping the thermodynamic-related precipitation almost intact (Fig. 4). Results from these last-millennium experiments confirm the critical role of a preceding El Niño in enhancing the EASM, against the well-known cooling and hydrological weakening effect of volcanic eruptions[13–16].

## Discussion
The fraction of El Niños occurring after volcanic eruptions in reconstructions is 41% for the study period from 1470 to 1999 AD (Fig. 2), or 44% if we expand the period to cover the last millennium from 901 to 1999 AD, which are both larger than the expected percentage (27%) for El Niño events without a volcanic effect. Despite the small sample sizes, i.e., 22 or 39 eruptions in the two periods, respectively, the results consistently suggest that tropical eruptions do increase the likelihood of an El Niño. This explains why the composite EASM rainfall after all eruptions is

increased in reconstructions. In the models, this count fraction is only 29% for PMIPs (Fig. 3), resulting in a feeble EASM rainfall increase in the all-eruption case. This less frequent occurrence of El Niño, against a robust equatorial Pacific westerly response to tropical volcanic eruptions, was argued to be caused by weak coupling between central Pacific precipitation and westerly anomalies in the PMIP3 and CESM models[48].

Although our ensemble analysis shows that a tropical eruption can increase the likelihood of an El Niño, individual paleoclimate reconstructions exhibit divergent responses[22,28,49,50]. We use the ensemble mean to maximize the signals, but the results might be affected by the proxy overlap among the 11 available ENSO indices. The divergent ENSO[26,27] and hydrology[47,51–54] responses, plus the time-dependent shifts of the Intertropical Convergence Zone (ITCZ)[55] to volcanic eruptions with asymmetric hemispheric distributions further complicate the volcano-EASM relationship quantification. Future study expanding to Northern or Southern Hemispheric eruptions should correct for SST variations associated with such ITCZ shifts.

To conclude, this work finds an east-west asymmetric monsoonal-ocean response to the zonally symmetric volcanic forcing, highlighting the delayed oceanic responses to volcanic eruptions in exciting EASM precipitation changes. A precipitation increase due to wind convergence induced by the enhanced western North Pacific high overwrites the precipitation reduction due to the thermodynamically induced moisture deficiency. The tropical Pacific plays a critical role in modulating the climate system response to external volcanic forcing, i.e., increasing the likelihood of an El Niño and enhanced post-El Niño warm pool air-sea interaction. Similar cases are found in the Nuclear Niño[34] or the northern Eurasia winter warming responses to stratospheric soot and sulfate aerosols injections during volcanic scenarios[56]. Results obtained from this work demonstrate the important role and complexity of the coupled atmosphere-ocean dynamics in Pacific, and shed light on the potential impact of volcanic eruptions on EASM hydroclimate.

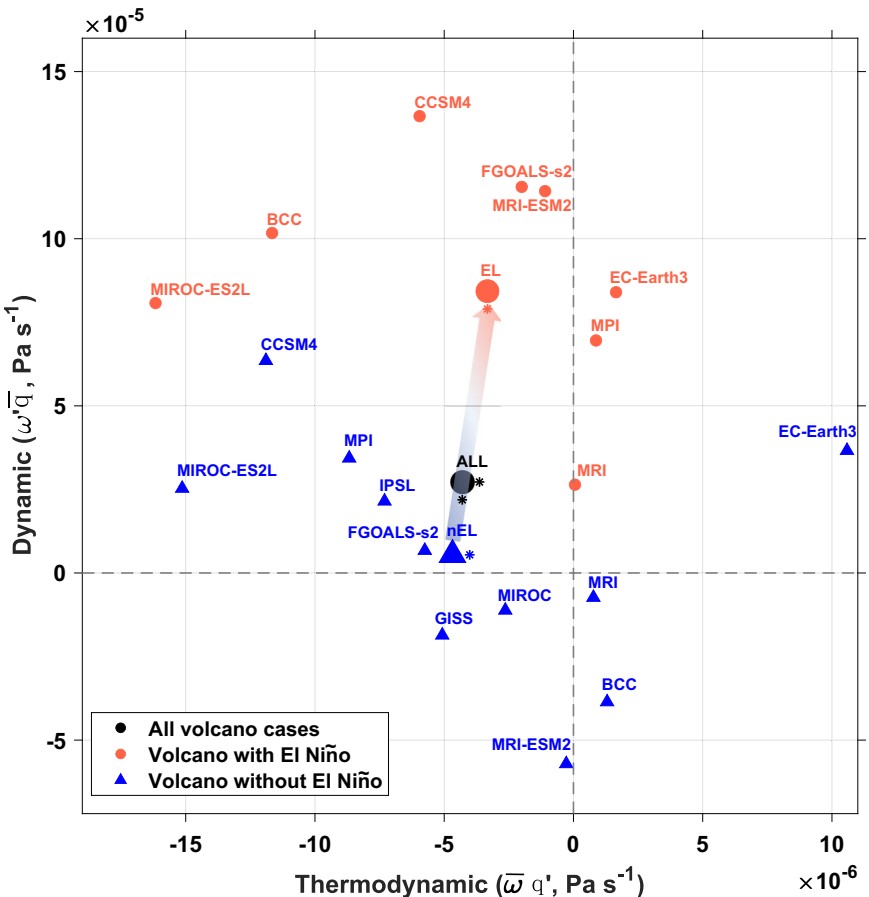

**Fig. 4 Simulated dynamic and thermodynamic responses of EASM to tropical eruptions modulated by El Niño.** Plotted are the composite dynamic part ($\omega'\bar{q}$) versus thermodynamic part ($\bar{\omega}q'$) of post-eruption EASM precipitation changes after all 78 eruptions in 11 PMIP last millennium simulations from 1470 to 1849 AD (All, dark dot), as well as those of the 21 eruptions with (EL, orange dot) and 57 eruptions without (nEL, blue triangle) El Niño responses in the first boreal winter after eruptions. CSIRO and HadCM last millennium simulations are not included due to the lack of vertical velocity. Asterisks to the right and bottom of the symbols denote the thermodynamic and dynamic anomalies significant at the 90% confidence level, respectively. The arrow indicates a change from the ensemble mean for eruptions without El Niño responses to those with El Niño responses. Results for each model are also shown by the small dots and triangles. Significant tests have not been performed for single models due to the small sample size of El Niño events. This figure was created using MATLAB 2020a (http://www.mathworks.com).

## Methods

**Proxy data**. The proxy data we analyzed mainly include a long-term reconstruction of Asian summer precipitation[36] and 11 reconstructed ENSO indices (Supplementary Table 1). The Feng et al.[36] data are a gridded (0.5° × 0.5°) reconstruction of annual May–September precipitation from 1470 to 1999 AD over the whole Asian continent and are mainly based on 500-year historical documentary records, tree-ring data, ice-core records, and a few long-term instrumental data series. A regional empirical orthogonal function method is utilized to increase the signal-to-noise ratio. Two other gridded reconstructions of precipitation based on point-by-point regression methods are also analyzed in parallel, serving as confirmation of the results: a 530-yr multi-proxy May–September precipitation reconstruction over China (0.5° × 0.5°) based on the tree-ring chronologies and drought/flood indices from 1470 to 2000 AD[37], and an annual June-August rainfall reconstruction over the Asian land region (2° × 2°) generated by merging 453 tree-ring width chronologies and 71 historical documentary records from 1470 to 2013[38].

To test the potential role of a preceding El Niño on volcanic-induced EASM rainfall change, the ensemble mean of 11 available ENSO indices is reconstructed and applied, following the method used before[26] except adding one new proxy[49]. All of the 11 ENSO indices first go through a 9-yr Lanczos high-pass filter[57] to isolate the ENSO signal, and then are normalized according to their own standard deviations. The ensemble mean is utilized to remove proxy uncertainty. Since these 11 indices cover different periods, with the longest extending from 900 to 2002 AD and the shortest from 1706 to 1997 AD, the ensemble mean for each year from 1470 to 1999 AD is calculated as the average of available indices. The ensemble mean with any individual index removed is still highly correlated (above the 99% confidence level), with a minimum correlation coefficient of 0.95, to the ensemble mean of all 11 indices. The resulting ensemble mean has a high correlation with the instrumental December–February Niño3.4 index of 0.83 (at the 99% confidence

level) for the period of 1871–1999, based on the averaged (5°N–5°S, 120°–170°W) SST anomaly from Hadley Centre Ice and SST version 1 (HadISST1)[58].

All 22 tropical eruptions from the Sigl et al.[35] volcanic reconstructions during the period of 1470–1999 AD, when Asian precipitation proxies are available, are used for the composite analysis. We define the peak aerosol loading year as the eruption year 0 (Supplementary Table 2).

**Last-millennium simulations**. All of the 10 last-millennium climate simulations in the PMIP3 and three simulations in the PMIP4 are analyzed to understand the underlying mechanisms of observed EASM response to large volcanic eruptions (Supplementary Table 3). Additional outputs from CESM's 10 last-millennium all-forcing simulations are also investigated, due to CESM's good performance in simulating ENSO seasonality, amplitude, frequency, and teleconnection[59]. Since different volcanic forcing reconstructions are used in the last-millennium simulations from PMIP models and CESM (Supplementary Table 3), the accuracy of the eruption date and strength in each last-millennium simulation is determined according to the volcanic forcing used, and year 0 denotes the year with maximum annual-mean stratospheric sulfate aerosol injection for each eruption, consistent with the using of peak aerosol loading year in reconstruction analysis[35]. A tropical eruption is defined when it has an aerosol density or aerosol optical depth in both hemispheres, following the ice core-based forcing reconstructions[60,61]. Thus, tropical eruptions have their maximum aerosol density or optical depth in the tropics (20°S–20°N)[47]. Since the PMIP simulations only cover the pre-industrial period before 1850, we have 92 eruption events in the 13 PMIP model simulations for a period of 1470–1849 AD and 228 events for the whole last millennium of 850–1849 AD, listed in Supplementary Table 4. In the 10 full forcing ensembles of CESM we have 90 eruption events for the period of 1470–1999 AD when the Asian precipitation reconstructions are available.

**Composite and significance**. A superposed epoch analysis[22] is used to evaluate the influence of explosive tropical volcanoes on global cooling and related EASM-ocean interaction. To isolate the climate responses to an eruption from the background noise, we obtain the anomaly by removing the climatology of the five years preceding the eruption. We apply the superposed epoch analysis to the reconstructed EASM precipitation and Niño3.4 indices based on the 22 reconstructed tropical eruptions, and internal variability is expected to be filtered out. A superposed epoch analysis is also performed on multi-model simulations from PMIP3 and PMIP4 and on multi-ensemble simulations from CESM. The significance of the superposed epoch analysis results for the 11-year window of five years before and six years after each eruption is calculated by the bootstrapped resampling method with 10,000 random draws from the full 500-year (i.e., 1470–1990 AD) study period[22].

**Climate indices and variability**. The Niño3.4 index is defined as the area-averaged SST anomaly over the area (5°N–5°S, 120°–170°W). To use the same standard for selecting an El Niño event list among reconstructions with different scales, an El Niño event is defined when the boreal winter Niño3.4 index is >0.5 standard deviations in both reconstructions and simulations. A non-volcanic El Niño is defined as when it occurs in a normal year without an eruption. Using different thresholds to select El Niño events does not change our results qualitatively. Boreal summer is defined as May–September, and boreal winter, as December–February, consistent with those used for reconstructions. The EASM region is defined as the region over (25°–34°N, 106°–122°E) in both reconstructions and simulations. Based on the simulated western North Pacific subtropical high, the EASM circulation index is defined as the 850-hPa zonal wind difference between the northern (22°–32°N, 130°–160°E) and southern (9°–19°N, 120°–150°E) regions, a little different from the original index between northern (22.5°–32.5°N, 110°–140°E) and southern (5°–15°N, 90°–130°E) regions from instrumental observations[62].

The monsoon precipitation index $P$ can be represented by moisture convergence[47,63]:

$$P' = -(\bar{q}\omega' + q'\bar{\omega} + q'\omega') \tag{1}$$

where $q$ denotes surface specific humidity and $\omega$ is the pressure velocity at 500 hPa. The overbar denotes the mean state before the eruption, while the prime, the anomaly after the eruption. $\bar{q}\omega'$ and $q'\bar{\omega}$ denote the dynamic and thermodynamic-related precipitation changes, respectively. $q'\omega'$ represents the nonlinear feedback and is usually much smaller than the dynamic or thermodynamic counterparts.

## Data availability
All data used in this study were obtained from publicly available sources. The PMIP model outputs are distributed by the Earth System Grid Federation (ESGF) at https://esgf-node.llnl.gov/projects/esgf-llnl/ (see Supplementary Table 3 for model details and references). Output from the CESM Last Millennium Ensemble (CESM-LME) can be downloaded at https://www.earthsystemgrid.org/dataset/ucar.cgd.ccsm4.cesmLME.html. Proxy reconstruction datasets are available at the National Centers of Environmental Information via https://www.ncdc.noaa.gov/data-access/paleoclimatology-data/datasets/climate-reconstruction.

## Code availability
The code used to create Figs. 1–4 in this study is available on https://zenodo.org/record/6503745. Other analytical scripts are available from the authors upon request. All of the figures are created by the authors using MATLAB 2020a (http://www.mathworks.com/) or NCAR Command Language (NCL; http://www.ncl.ucar.edu/). The atlas imbedded in the figures are the by-default atlas from MATLAB or NCL.

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

## Acknowledgements

F.L. is supported by the National Natural Science Foundation of China (Grant 41975107). C.G is supported by the National Natural Science Foundation of China (Grant 41875092). A.R. is supported by the U.S. National Science Foundation grant AGS-2017113. B.W. acknowledges support from U.S. National Science Foundation grant 2025057. G.H. is supported by the National Natural Science Foundation of China (Grant 41721004, 41831175).

## Author contributions

F.L., C.G., and B.W. conceptualized and led the work. F.L., C.G., and J.C. contributed to data analysis including validation and interpretation of the results. F.L., C.G., A.R., and B.W. wrote the manuscript, J.L., X.Z., G.H., and W.D. reviewed and edited the manuscript.

## Competing interests

The authors declare no competing interests.
