## [Peer Review File · Nature Communications]

Tropical volcanism enhanced the East Asian summer monsoon during the last millenniumREVIEWER COMMENTS

Reviewer #1 (Remarks to the Author):

The manuscript addresses the impact of tropical volcanism on the East Asian Summer Monsoon. The conclusions are well supported and interesting and I couldn't find any major flaw. However, the topic has been already addressed in other papers (Cui et al., 2014 <https://link.springer.com/article/10.1007/s00376-014-3239-8>; and other papers cited in the article) although with controversial/opposite results. Therefore, I am not sure the results are of enough general interest for Nature Communications and I feel the article may be better suited for a specific journal.

The article is clear and well written, and the figures are also very well done. Please increase the fontsize of the numbers in the colorscales.

Reviewer #2 (Remarks to the Author):

Overview

This study by Liu et al. investigates the relationship between tropical volcanism and the East Asian Summer Monsoon (EASM) rainfall. By conducting superposed epoch analysis on a gridded reconstruction of EASM rainfall (Feng et al., 2013), the authors find that the occurrence of post-eruption El Niño can enhance the EASM rainfall, in contrast to the previous theory that volcanism will suppress the EASM precipitation. This finding is then confirmed with climate simulations, and mechanisms are studied based on these simulations.

Overall, I find the topic of this study of importance and interest to the community, the manuscript in good quality, and this study helps to form a better understanding of the hydrological impact of post-eruption El Niño events. That said, some clarifications and modifications are needed, which I list below. If those could be addressed, I recommend the work be accepted for publication.

Details

- The title: Fig. 2b of this study has indicated that, for cases without a preceding El Niño, the EASM rainfall is not enhanced after eruptions. Therefore, the current title is misleading to some extent. It might be more accurate to say "Post-eruption El Niño Enhanced the East Asian Summer Monsoon during the Last Millennium" or something similar, as the causal linkage between tropical volcanism and El Niño is not conclusive as the authors have noted in L181-183.

- L166-168: It should be noted to the readers that the fraction is a small sample statistics ($n=22$), and it was suggested (Self et al., 1997; Lehner et al., 2016) that El Niño events were already underway before the eruptions began regarding El Chichon (1982) and Pinatubo (1991), implying a potential of coincidence between ENSO activity and volcanism for these events.

- L183-184 and Fig. S3: It might not be appropriate to construct the ensemble mean of the multiple ENSO reconstructions used here, as many of them are not independent. For instance, the first principal component of NADA is used in both Li et al. (2011) & (2013), and McGregor et al. (2010) is a composite of several previous reconstructions, including Stahle et al. (1998), Mann et al. (2000), Cook et al. (2008) and Braganza et al. (2009), potentially counting common predictors multiple times.

- L184-187: The logic here is somewhat hard to understand. Fig. 2 of this study shows that the EASM precipitation is enhanced with El Niño (9 cases) and not enhanced without El Niño (13 cases). This alone suggests that volcanism is not the leading factor here, but El Niño is. Fig. 1 shows that EASM rainfall is overall enhanced considering all 22 cases, which only implies that the 9 cases overwhelm the 13 cases. Since the superposed epoch analysis (SEA) involves the calculation of the composite mean, and that the sample size is small here ($n=22$), the significantly large amplitude in the 9 cases (or even in some of the 9 cases) can likely dominate the composite mean. I don't see the connection to "increased post-eruption El Niño occurrence" here, and clarifications are needed.

- Figs. 2a, 3c, 3f: In the significance test for the cases with preceding El Niño, what is the reference pool? As mentioned above, it appears that volcanism is not the leading factor, but El Niño is. Therefore, to investigate the impact of volcanism, we need additional tests to tell if post-eruption El Niño events and normal El Niño events (during non-volcanic years) contribute differently, in which case we use the normal El Niño events as the reference pool. Details about the test should be described in the Methods section.

Reviewer #3 (Remarks to the Author):

Review of

Tropical Volcanism Enhanced the East Asian Summer Monsoon during the Last Millennium

Fei Liu and co-authors

General

The authors study the response of the EASM to (tropical) volcanic eruptions. They provide paleo-evidence that EASM intensity increases following eruptions, and following Niño events. Since Niño events are more prevalent following volcanic eruptions, EASM generally increases following eruptions because the Niño response overwhelms the reduction in precipitation, seen throughout the tropics, due to the

global cooling that follows eruptions.

The authors make a compelling case for the Niño-EASM link and for the increased prevalence of Niño events following eruptions, based on reconstructions and model simulations. The paper is generally well organized, though I think the narrative of the paper can be made clearer. Overall, the paper provides a meaningful contribution to our understanding of EASM variability. However, I found some limitations in the methodology and interpretation of the results. I would therefore recommend accepting the paper, following the comments provided below.

Major comments

1. The ending of the paper is somewhat confusing to me. The argumentation in the early parts of the paper supports the claim that Niño events following eruptions lead to enhanced precipitation in EASM, despite general tropical cooling and reduced precipitation. But then, the concluding sentence of the paper inverts the argument. It points to evidence of increased precipitation following eruptions as indication of increased occurrence of Niño events following eruptions. The authors should be clearer about the key takeaway messages of this paper.
2. The nature of the ENSO-like response following eruptions is not clear. In the ensemble means shown in Fig. 3 for CESM LME, even the non-Niño events show a relative warming of the eastern vs. western Pacific (on top of a tropical mean cooling trend). An Niño-like response is therefore generally seen following all events, even if the amplitude of the relevant index does not go beyond the cutoff to qualify as Niño events. An alternate interpretation of the results is therefore, that, consistent with the dynamic thermostat mechanism, a Niño-like response is seen after all eruptions, in response to the global cooling. The strongest Niño response

leads to EASM enhancement while the weakest Niño response is not enough to counter the reduced precipitation caused by global cooling.

3. In reconstruction (Fig. S3) we see increased precipitation in the first winter following eruptions and then a strong reduction in precipitation at year 2-3. This is consistent with the typical Niña-like response ~3 years after eruptions (Stevenson et al. 2016, Erez and Adam 2021). This may therefore support the ENSO-EASM link demonstrated by the authors, whereby the Niño-like response amplifies precipitation in EASM and the Niña-like response inhibits EASM precipitation. However, this also casts some doubt on the title of the paper because the Niña-like response may balance the Niño-like response, with no net EASM enhancement in response to tropical volcanism.
4. In a previous paper the lead author and co-authors showed that non-tropical eruptions weaken the monsoon in the eruption hemisphere. This paper focuses on tropical eruptions. However, the following question needs to be addressed: is the Niño enhancement following tropical eruptions also expected following asymmetric eruptions? More specifically, is this Niño enhancement sufficient to overcome the weakening of the EASM following NH eruptions? Even if the authors cannot answer this question in the present analysis, they should present it as a possible limitation of the main finding.

Minor comments

1. Note that the authors' definition of tropical eruptions differs from other works who defined "tropical eruptions" based on inter-hemispheric symmetry rather than the location of the peak.
2. In Fig. 3c, why are we not seeing Niño-like warming in the eastern equatorial Pacific, as in Fig. 3f?

Comments by line

- 50 "...southwesterly monsoon over the northwest flank of the western North Pacific subtropical high..." is a mouthful. First, what is a "southwesterly monsoon"? Second, please try and better communicate this sentence. Perhaps by breaking it into parts.
- 52 midlatitude westerlies
- 62 monsoon → EASM. You are addressing a specific region, not the general response of monsoons.
- 69 Note the recent publication by Erez and Adam (2021), who discuss the relation of ITCZ shifts to ENSO following volcanos.
- 74 of volcanic → by volcanic
- 76 examine

94 “suggesting quick monsoon-ocean feedback” is a bit speculative at this stage of the paper.

120-125 I have no idea what the authors are trying to say in this very long sentence.

146-148 This sentence is confusing. The reduction is because of the eruptions, not the preceding El Niño.

171-173 and 181-183 Erez and Adam (2021) point to the sensitivity to the choice of ENSO index and also suggest that when you correct for SST variations associated with ITCZ shifts, the appearance of Niño events after eruptions becomes more pronounced.

246 please quantify what you mean by “in the tropics”.

277-278 The said equation and decomposition to dynamic and thermodynamic components does not appear the associated reference (Liu et al. 2016, reference 45 in the text)

References

Erez, M., & Adam, O., 2021: Energetic Constraints on the Time-Dependent Response of the ITCZ to Volcanic Eruptions, *Journal of Climate*, **34**(24), 9989-10006

We would like to thank the reviewers for their positive evaluations of our work, and for the constructive comments. Below we cite the reviewer comments in *blue*, with our responses in black.

Reviewer #1 (Remarks to the Author):

1. The manuscript addresses the impact of tropical volcanism on the East Asian Summer Monsoon. The conclusions are well supported and interesting and I couldn't find any major flaw. However, the topic has been already addressed in other papers (Cui et al., 2014 <https://link.springer.com/article/10.1007/s00376-014-3239-8>; and other papers cited in the article) although with controversial/opposite results. Therefore, I am not sure the results are of enough general interest for Nature Communications and I feel the article may be better suited for a specific journal.

Reply 1:

We acknowledge that the impact of tropical volcanism on the East Asian Summer Monsoon (EASM) has been studied using observation, proxy, or model simulations in previous studies as the reviewer mentioned. This study is different from the previous ones and merits consideration for Nature Communications because: (1) we demonstrate, contrary to the well-known hydrological weakening theory of volcanic eruptions (which is supported in Cui et al. 2014), that a volcano-induced El Niño and the associated stronger than non-volcanic El Niño warm pool air-sea interactions intensify EASM precipitation; (2) the results are drawn from the most comprehensive proxy observations and multi-model analyses that are currently available.

2. The article is clear and well written, and the figures are also very well done. Please increase the font size of the numbers in the color scales.

Reply 2:

The font size for Fig. 3 has been increased.

Reviewer #2

Overview: This study by Liu et al. investigates the relationship between tropical volcanism and the East Asian Summer Monsoon (EASM) rainfall. By conducting superposed epoch analysis on a gridded reconstruction of EASM rainfall (Feng et al., 2013), the authors find that the occurrence of post-eruption El Niño can enhance the EASM rainfall, in contrast to the previous theory that volcanism will suppress the EASM precipitation. This finding is then confirmed with climate simulations, and mechanisms are studied based on these simulations.

Overall, I find the topic of this study of importance and interest to the community, the manuscript in good quality, and this study helps to form a better understanding of the hydrological impact of post-eruption El Niño events. That said, some clarifications and modifications are needed, which I list below. If those could be addressed, I recommend the work be accepted for publication.

Details

- 1. The title: Fig. 2b of this study has indicated that, for cases without a preceding El Niño, the EASM rainfall is not enhanced after eruptions. Therefore, the current title is misleading to some extent. It might be more accurate to say "Post-eruption El Niño Enhanced the East Asian Summer Monsoon during the Last Millennium" or something similar, as the causal linkage between tropical volcanism and El Niño is not conclusive as the authors have noted in L181-183.*

Reply 1:

The **composite** EASM rainfall increases after all volcano eruptions, suggesting a surprising fact supporting the current title.

It has been well established that El Niño can strengthen EASM (Meiyu) rainfall after its mature phase (e.g., Wang et al. 2000). Therefore, if we say “post-eruption El Niño enhanced EASM rainfall”, it would be a trivial statement.

The novelty of the present work is that EASM rainfall is overall enhanced after volcano eruptions regardless of their cooling effects. The fundamental reason is that tropical volcanic eruption-induced radiative cooling can instigate atmosphere-ocean interaction in the Pacific, resulting in a post-eruption El Niño or an El Niño-like cooling (if the El Niño is not fully developed). In both cases, the equatorial SST gradients were reduced (like in an El Niño). It is the SST gradients that favor increasing EASM rainfall against the eruption-cooling effects.

There are three pieces of evidence that support our hypothesis. First, tropical eruptions significantly increase the likelihood of El Niño occurrence. The fraction of El Niños occurring after volcanic eruptions in reconstructions is 41% for the study period from 1470 to 1999 AD (Fig. 2), or 44% if we expand the period to cover the last millennium from 901 to 1999 AD, which are both larger than the expected percentage (27%) for El Niño events without the volcanic effect. Second, the post-eruption El Niño shows a stronger El Niño-EASM relationship than the normal El Niño without the volcanic effect (revised Figs. 2c and 3d). Third, even for the post-eruption cases without a fully developed El Niño, the Pacific still features an El Niño-like cooling. The El Niño-like cooling can weaken the eruption-induced drying influence on EASM. This explains why the nine post-eruption El Niño cases can overwhelm the 13 no-post-

eruption El Niño cases, resulting in the composite (overall) increase of the EASM rainfall. In revision, we have implemented and elaborated reasons #2 and #3 in detail.

- L166-168: It should be noted to the readers that the fraction is a small sample statistics (n=22), and it was suggested (Self et al., 1997; Lehner et al., 2016) that El Niño events were already underway before the eruptions began regarding El Chichon (1982) and Pinatubo (1991), implying a potential of coincidence between ENSO activity and volcanism for these events.*

Reply 2:

We've put the notation in the revision. We also repeat the examination with 39 eruptions covering the last millennium from 901 to 1999 AD, and the results further support the conclusion that tropical eruptions do increase the likelihood of an El Niño. The relevant revision is copied below:

L173-178: *“The fraction of El Niños occurring after volcanic eruptions in reconstructions is 41% for the study period from 1470 to 1999 AD (Fig. 2), or 44% if we expand the period to cover the last millennium from 901 to 1999 AD, which are both larger than the expected percentage (27%) for El Niño events without the volcanic effect. Despite the small sample sizes, i.e., 22 or 39 eruptions in the two periods, respectively, the results consistently suggest that tropical eruptions do increase the likelihood of an El Niño.”*

The 1982 and 1991 El Niños were initiated before the two recent eruptions, but they would not have been as strong as observed if there was no impact of volcanic eruptions, according to the operational prediction systems which do not include volcanic impacts as forcing (personal communication). Therefore, despite the potential coincidence between El Niño and eruption in these two (and possibly other) cases, the latter does intensify the El Niño events.

- L183-184 and Fig. S3: It might not be appropriate to construct the ensemble mean of the multiple ENSO reconstructions used here, as many of them are not independent. For instance, the first principal component of NADA is used in both Li et al. (2011) & (2013), and McGregor et al. (2010) is a composite of several previous reconstructions, including Stahle et al. (1998), Mann et al. (2000), Cook et al. (2008) and Braganza et al. (2009), potentially counting common predictors multiple times.*

Reply 3:

We acknowledge the overlap in the proxies among the reconstructions and add a brief discussion in the revised text as copied below. To the best of our knowledge, ENSO index reconstructions depend on both the proxy and methodology used. Our pairwise correlation analysis shows complex patterns of correlation among the indices. For example, the correlation between Li et al. 2011 (EN1) and 2013 (EN5) is low (Table A), although the NADA index was used in both reconstructions. The independent coral-based reconstruction of Dee et al. 2020 (EN11) is significantly different from the others, but it only covers 9 eruption events due to the short data length.

Given the existing issues in the available ENSO reconstruction, we use the ensemble mean as a simple way to account for as many signals as possible and meanwhile to minimize the uncertainties. In the revision, we conduct additional tests by removing an individual index to confirm that the ensemble mean is not controlled by any one dataset.

L184-187: “Although our ensemble analysis shows that a tropical eruption can increase the likelihood of an El Niño, individual paleoclimate reconstructions exhibit divergent responses (Adams et al. 2003; Dee et al. 2020; Robock 2020). We use the ensemble mean to maximize the signals, but the results might be affected by the proxy overlap among the 11 available ENSO indices.”

L237-240: “The ensemble mean with any individual index removed is still highly correlated (above the 99% confidence level), with a minimum correlation coefficient of 0.95, to the ensemble mean of all 11 indices.”

Table A. Correlation coefficients among 11 different ENSO reconstructions.

	EN2	EN3	EN4	EN5	EN6	EN7	EN8	EN9	EN10	EN11
EN1	0.69	0.74	0.30	0.48	0.59	0.10	0.74	0.52	0.68	-0.07
EN2		0.83	0.52	0.55	0.67	0.12	0.85	0.60	0.75	-0.28
EN3			0.45	0.56	0.67	0.15	0.87	0.59	0.79	-0.04
EN4				0.48	0.54	0.24	0.75	0.62	0.63	-0.21
EN5					0.61	0.25	0.69	0.50	0.63	0.02
EN6						0.09	0.83	0.59	0.67	-0.31
EN7							0.26	0.19	0.28	0.58
EN8								0.78	0.89	-0.12
EN9									0.68	-0.17
EN10										-0.01

4. L184-187: The logic here is somewhat hard to understand. Fig. 2 of this study shows that the EASM precipitation is enhanced with El Niño (9 cases) and not enhanced without El Niño (13 cases). This alone suggests that volcanism is not the leading factor here, but El Niño is. Fig. 1 shows that EASM rainfall is overall enhanced considering all 22 cases, which only implies that the 9 cases overwhelm the 13 cases. Since the superposed epoch analysis (SEA) involves the calculation of the composite mean, and that the sample size is small here ($n=22$), the significantly large amplitude in the 9 cases (or even in some of the 9 cases) can likely dominate the composite mean. I don't see the connection to "increased post-eruption El Niño occurrence" here, and clarifications are needed.

Reply 4:

As discussed in our replies to comments #1 and #2, tropical volcanic eruptions do enhance the likelihood of an El Niño in our analysis of either 22 events during 1470-1999 AD or 39 events during 901-1999 AD. Moreover, those 13 cases that are not followed by El Niño events also exhibit an El Niño-like Pacific cooling which weakens the eruption-induced drying influence on EASM.

Nevertheless, we do acknowledge that the logic in the original L184-187 was not made clear. In the revision, we have removed the seemingly circular argument, added the additional results with 39 eruption cases, and modified the relevant discussion to make the takeaway message more clearly.

5. *Figs. 2a, 3c, 3f: In the significance test for the cases with preceding El Niño, what is the reference pool? As mentioned above, it appears that volcanism is not the leading factor, but El Niño is. Therefore, to investigate the impact of volcanism, we need additional tests to tell if post-eruption El Niño events and normal El Niño events (during non-volcanic years) contribute differently, in which case we use the normal El Niño events as the reference pool. Details about the test should be described in the Methods section.*

Reply 5:

Because we show anomalies with respect to the climatology, the reference pool for the significance test is the full 500-year period (i.e., 1470-1990 AD) from which we performed the bootstrapped resampling method with 10,000 random draws.

Thank you for raising this issue. In the revision, we followed your suggestion and compared the post-eruption El Niño and normal El Niño in both reconstructions and simulations. The results show that tropical eruptions do intensify the post-El Niño air-sea interaction and significantly enhance the El Niño-EASM relationship (Figs. 2a&c and Figs. 3c&d).

Reviewer #3

General The authors study the response of the EASM to (tropical) volcanic eruptions. They provide paleo-evidence that EASM intensity increases following eruptions, and following Niño events. Since Niño events are more prevalent following volcanic eruptions, EASM generally increases following eruptions because the Niño response overwhelms the reduction in precipitation, seen throughout the tropics, due to the global cooling that follows eruptions.

The authors make a compelling case for the Niño-EASM link and for the increased prevalence of Niño events following eruptions, based on reconstructions and model simulations. The paper is generally well organized, though I think the narrative of the paper can be made clearer. Overall, the paper provides a meaningful contribution to our understanding of EASM variability. However, I found some limitations in the methodology and interpretation of the results. I would therefore recommend accepting the paper, following the comments provided below.

a. Major comments 1. The ending of the paper is somewhat confusing to me. The argumentation in the early parts of the paper supports the claim that Niño events following eruptions lead to enhanced precipitation in EASM, despite general tropical cooling and reduced precipitation. But then, the concluding sentence of the paper inverts the argument. It points to evidence of increased precipitation following eruptions as indication of increased occurrence of Niño events following eruptions. The authors should be clearer about the key takeaway messages of this paper.

Reply a: We have revised the discussion by removing the seemingly circular argument and clarified the key takeaway message. The key message is that tropical eruptions increase the likelihood of El Niño-like responses and strengthen the El Niño-EASM relationship.

b. The nature of the ENSO-like response following eruptions is not clear. In the ensemble means shown in Fig. 3 for CESM LME, even the non-Niño events show a relative warming of the eastern vs. western Pacific (on top of a tropical mean cooling trend). An Niño-like response is therefore generally seen following all events, even if the amplitude of the relevant index does not go beyond the cutoff to qualify as Niño events. An alternate interpretation of the results is therefore, that, consistent with the dynamic thermostat mechanism, a Niño-like response is seen after all eruptions, in response to the global cooling. The strongest Niño response leads to EASM enhancement while the weakest Niño response is not enough to counter the reduced precipitation caused by global cooling.

Reply b:

Thanks for your observation and insightful comment. We have revised the text, following your suggestion, to interpret our results more constructively and clearly.

L117-139: *“In the first post-eruption summer (Fig. 3b), the composite for 68 out of the 92 events without a preceding El Niño mainly exhibits a direct cooling and drying volcanic effect over most parts of the globe. The relative El Niño signal is simulated and the reduced zonal SST gradient across the Pacific weakly enhances the Pacific*

High, represented by an anticyclonic anomaly mainly over the South China Sea. Studies have established that the SST gradients across the tropical Pacific strongly influence global rainfall (Ropelewski and Halpert 1987). In the presence of a preceding El Niño, the model results show increased post-eruption summer precipitation in the EASM region, and the most significant enhancement is located over the middle-lower reach of Yangtze River, driven by the convergence and moisture advection of southwesterly wind anomalies of the enhanced Pacific High and extratropical northerly anomalies, albeit not significant, downstream of the Tibetan Plateau (Fig. 3c). Thus, El Niño-like response in non-El Niño events tends to offset the cooling-induced dry monsoon over East Asia, resulting in an insignificant decrease in the EASM rainfall. Thus, the post-eruption El Niño cases can overwhelm the no-post-eruption El Niño cases, resulting in overall increase of the EASM rainfall.

Eruption-induced radiative cooling leads to an El Niño in the post-eruption winter or an El Niño-like response in the following summer when the El Niño is not fully developed. The responses is determined by the coupled atmosphere-land-ocean dynamics. The decrease in SST gradient, i.e., relative warmer eastern Pacific, can be initiated by eruption-induced global cooling through ocean dynamic thermostat mechanism (Clement et al. 1996; Mann et al. 2005) and land-sea thermal contrast (Khodri et al. 2017; Ohba et al. 2013). This decreased SST gradient gives rise to a weakened pressure gradient and hence weaker easterly winds and Walker circulation, which in turn reduce the SST gradient, a mechanism known as “the Bjerknes feedback” (Bjerknes 1969)”

c. *In reconstruction (Fig. S3) we see increased precipitation in the first winter following eruptions and then a strong reduction in precipitation at year 2-3. This is consistent with the typical Niña-like response ~3 years after eruptions (Stevenson et al. 2016, Erez and Adam 2021). This may therefore support the ENSO-EASM link demonstrated by the authors, whereby the Niño-like response amplifies precipitation in EASM and the Niña-like response inhibits EASM precipitation. However, this also casts some doubt on the title of the paper because the Niña-like response may balance the Niño-like response, with no net EASM enhancement in response to tropical volcanism.*

Reply c: Fig. S3 (Fig. S5 in the revision) shows the reconstructed ENSO response to tropical eruptions. Our analysis does not show negative precipitation responses in the third summer (Fig. 1b), and rather a La Niña-like response is observed (Fig. S5). This is probably due to the asymmetric ENSO-EASM teleconnection between its warm and cold phases, and the precipitation is not suppressed following a La Niña. Wang et al. (2020) have shown that the Asian precipitation response to ENSO exhibits an asymmetry between El Niño and La Niña events, mainly over the East Asian region (their Figs. 8). The Asian precipitation–La Niña relationship is more variable than the Asian precipitation–El Niño relationship, especially for minor La Niña events. Therefore, we prefer to keep our original title.

Wang, B., X. Luo, and J. Liu, 2020: How Robust is the Asian Precipitation–ENSO Relationship during the Industrial Warming period (1901-2017)? *J. Climate*, 33, 2779-2792.

d. In a previous paper the lead author and co-authors showed that non-tropical eruptions weaken the monsoon in the eruption hemisphere. This paper focuses on tropical eruptions. However, the following question needs to be addressed: is the Niño enhancement following tropical eruptions also expected following asymmetric eruptions? More specifically, is this Niño enhancement sufficient to overcome the weakening of the EASM following NH eruptions? Even if the authors cannot answer this question in the present analysis, they should present it as a possible limitation of the main finding.

Reply d:

In one of our previous reconstruction analyses (Liu et al. 2018CD) we did find that, the Northern Hemispheric eruptions tend to significantly increase the likelihood of an El Niño in the second post-eruption winter, and the model results suggest that it is because the westerly wind anomalies could be excited over equatorial Pacific due to the equatorward migration of the ITCZ. It will be interesting to explore whether the EASM precipitation will be increased by this El Niño enhancement effect, against the Northern Hemispheric cooling in the second summer after the NH eruptions, in a future study.

In the revision, we briefly discuss the possible EASM response to asymmetric eruptions combining results from Liu et al. (2018) and Erez and Adam (2021). The latter found that the shifts of ITCZ could affect SSTs especially in the Pacific, leading to enhanced equatorial zonal SST gradients following SH events and weakened gradients following NH events. These variations may superimpose on concomitant ENSO-like variations, complicating the EASM precipitation responses. We therefore propose it as a future study.

L187-191: “*The divergent ENSO(Liu et al. 2018; McGregor et al. 2020) and hydrology (Colose et al. 2016; Gao et al. 2021; Liu et al. 2016; Stevenson et al. 2016; Zuo et al. 2019) responses, plus the time-dependent shifts of the ITCZ (Erez and Adam 2021) to volcanic eruptions with asymmetric hemispheric distributions further complicate the relationship quantification. Future study expanding to Northern or Southern Hemispheric eruptions should correct for SST variations associated with such ITCZ shifts*”.

Minor comments

1. Note that the authors’ definition of tropical eruptions differs from other works who defined “tropical eruptions” based on inter-hemispheric symmetry rather than the location of the peak.

Reply 1: The reason we choose to use the peak location instead of inter-hemispheric symmetry as the definition of tropical eruptions is that, in the volcanic forcing used in PMIP3 and CESM (Gao et al., 2008 and Crowley et al., 2008) the datasets are presented with their aerosol density or aerosol optical depth in either the Northern or Southern

Hemisphere only, or in both as tropical eruptions. In other words, we choose to keep the original definition adopted by the volcanic reconstructions (Gao et al. 2008), and in the meantime avoid assigning asymmetric eruptions like 1982 El Chichón as a NH event.

In the revision we add a brief explanation on why we choose this method: **L258-261**, “A tropical eruption is defined when it has aerosol density or aerosol optical depth in both hemispheres, following the ice core-based forcing reconstructions (Crowley et al. 2008; Gao et al. 2008). Thus, tropical eruptions have their maximum aerosol density or optical depth in the tropics (20°S-20°N) (Liu et al. 2016).”

2. *In Fig. 3c, why are we not seeing Niño-like warming in the eastern equatorial Pacific, as in Fig. 3f?*

Reply 2:

Figures 3c and 3f show the SST anomalies in the summer after the El Niño peak phase (with Niño3.4 above 0.5 standard deviations). In the revision we added Supplementary Fig. S2 to show the SST anomalies in the first post-eruption winter, where we do see El Niño-like warming in the eastern equatorial Pacific.

Since we added one new panel to show the results of the non-volcanic El Niños in Fig. 3, the CESM results originally in Fig. 3 were moved to the supplementary.

Comments by line

3. 50 “...southwesterly monsoon over the northwest flank of the western North Pacific subtropical high...” is a mouthful. First, what is a “southwesterly monsoon”? Second, please try and better communicate this sentence. Perhaps by breaking it into parts.

Reply 3:

We have revised the statement to eliminate the ambiguity:

L49-51: “EASM rainfall is usually enhanced by moisture advection and convergence due to the southwesterly anomaly of the enhanced western North Pacific subtropical high.”

4. 52 midlatitude westerlies

Reply 4: It has been corrected.

5. 62 monsoon → EASM. You are addressing a specific region, not the general response of monsoons.

Reply 5:

It has been revised as suggested.

6. 69 Note the recent publication by Erez and Adam (2021), who discuss the relation of ITCZ shifts to ENSO following volcanos.

Reply 6:

Since the mechanism of equatorward migration of ITCZ mainly works for the high-latitude eruption-induced El Niño, we deleted it in our revision.

7. 74 of volcanic → by volcanic

Reply 7:

It has been corrected.

8. 76 examine

Reply 8:

It has been corrected.

9. 94 “suggesting quick monsoon-ocean feedback” is a bit speculative at this stage of the paper.

Reply 9:

We moved this argument to the next paragraph after we discussed the role of a preceding El Niño: **L96-99**, “Our multi-proxy analysis results thus demonstrate that explosive tropical eruptions tend to strengthen the El Niño-EASM relationship and increase EASM rainfall in the first subsequent boreal summer, suggesting quick monsoon-ocean feedback after large tropical eruptions.”

10. 120-125 I have no idea what the authors are trying to say in this very long sentence.

Reply 10:

In this sentence, we want to describe the hydroclimate response in EASM for the 68 volcanic events without El Niño, therefore provide a comparison reference for the 24 events with El Niño. In the revision, we have broken the long sentence into shorter ones as copied below:

L119-124, “The relative El Niño signal is simulated and the reduced zonal SST gradient across the Pacific weakly enhances the Pacific High, represented by an anticyclonic anomaly mainly over the South China Sea. Studies have established that the SST gradients across the tropical Pacific strongly influence global rainfall (Ropelewski and Halpert 1987). Thus, El Niño-like response in non-El Niño events tends to offset the cooling-induced dry monsoon over East Asia, resulting in an insignificant decrease in the EASM rainfall.”

11. 146-148 This sentence is confusing. The reduction is because of the eruptions, not the preceding El Niño.

Reply 11:

The precipitation reduction in South Asia is mainly due to the eruptions and partly because of El Niño. We have revised it as: **L152-154**, “After a volcano-induced El Niño, strong negative precipitation anomalies are simulated over South Asia, consistent with previous studies demonstrating the drying effect of large volcanic eruptions on the South Asian monsoon.”

12. 171-173 and 181-183 Erez and Adam (2021) point to the sensitivity to the choice of ENSO index and also suggest that when you correct for SST variations associated with ITCZ shifts, the appearance of Niño events after eruptions becomes more pronounced.

Reply 12:

In this paper we mainly focused on the effects of tropical eruptions, and the analysis shows tropical eruptions mainly weaken the ITCZ, with negligible meridional shifts. Nevertheless, the time-dependent shifts of the ITCZ and its influence on equatorial Pacific SST as pointed out in Erez and Adam (2021) merit careful consideration, when we expand the study to include NH or SH eruptions. Therefore, we have incorporated these results in the relevant discussions as copied below:

L184-191: *“Although our ensemble analysis shows that a tropical eruption can increase the likelihood of an El Niño, individual paleoclimate reconstructions exhibit divergent responses (Adams et al. 2003; Dee et al. 2020; Robock 2020). We use the ensemble mean to maximize the signals, but the results might be affected by the proxy overlap within the 11 available ENSO indices. The divergent ENSO(Liu et al. 2018; McGregor et al. 2020) and hydrology (Colose et al. 2016; Gao et al. 2021; Liu et al. 2016; Stevenson et al. 2016; Zuo et al. 2019) responses, plus the time-dependent shifts of the ITCZ (Erez and Adam 2021) to volcanic eruptions with asymmetric hemispheric distributions further complicate the relationship quantification. Future study expanding to Northern or Southern Hemispheric eruptions should correct for SST variations associated with such ITCZ shifts.”*

13. 246 please quantify what you mean by “in the tropics”.

Reply 13:

We defined the tropics in the revision: **L258-261**, *“A tropical eruption is defined when it has aerosol density or aerosol optical depth in both hemispheres, following the ice core-based forcing reconstructions (Crowley et al. 2008; Gao et al. 2008). Thus, tropical eruptions have their maximum aerosol density or optical depth in the tropics (20°S-20°N)(Liu et al. 2016).”*

14. 277-278 The said equation and decomposition to dynamic and thermodynamic components does not appear the associated reference (Liu et al. 2016, reference 45 in the text)

Reply 14:

This moisture convergence budget was shown by Fig. 8 of Liu et al. 2016 and Equation (3) of Hsu et al. 2012. We added the reference of Hsu et al. 2012 in the revision.

References

- Adams, J. B., M. E. Mann, and C. M. Ammann, 2003: Proxy evidence for an El Niño-like response to volcanic forcing. *Nature*, **426**, 274-278.
- Bjerknes, J., 1969: Atmospheric teleconnections from the equatorial Pacific. *Monthly weather review*, **97**, 163-172.
- Clement, A. C., R. Seager, M. A. Cane, and S. E. Zebiak, 1996: An Ocean Dynamical Thermostat. *Journal of Climate*, **9**, 2190-2196.

Colose, C., A. LeGrande, and M. Vuille, 2016: The influence of volcanic eruptions on the climate of tropical South America during the last millennium in an isotope-enabled general circulation model. *Climate of the Past*, **12**, 961-979.

Crowley, T. J., G. Zielinski, B. Vinther, R. Udisti, K. Kreutz, J. Cole-Dai, and E. Castellano, 2008: Volcanism and the Little Ice Age. *PAGES News*, **16**, 22-23.

Dee, S. G., K. M. Cobb, J. Emile-Geay, T. R. Ault, R. L. Edwards, H. Cheng, and C. D. Charles, 2020: No consistent ENSO response to volcanic forcing over the last millennium. *Science*, **367**, 1477-1481.

Erez, M., and O. Adam, 2021: Energetic Constraints on the Time-Dependent Response of the ITCZ to Volcanic Eruptions. *Journal of Climate*, **34**, 9989-10006.

Gao, C.-C., L.-S. Yang, and F. Liu, 2021: Hydroclimatic anomalies in China during the post-Laki years and the role of concurring El Niño. *Advances in Climate Change Research*, **12**, 187-198.

Gao, C., A. Robock, and C. Ammann, 2008: Volcanic forcing of climate over the past 1500 years: An improved ice core-based index for climate models. *Journal of Geophysical Research: Atmospheres*, **113**.

Khodri, M., and Coauthors, 2017: Tropical explosive volcanic eruptions can trigger El Niño by cooling tropical Africa. *Nature Communications*, **8**, 778.

Liu, F., J. Chai, B. Wang, J. Liu, X. Zhang, and Z. Wang, 2016: Global monsoon precipitation responses to large volcanic eruptions. *Scientific reports*, **6**, 1-11.

Liu, F., J. Li, B. Wang, J. Liu, T. Li, G. Huang, and Z. Wang, 2018: Divergent El Niño responses to volcanic eruptions at different latitudes over the past millennium. *Climate Dynamics*, **50**, 3799-3812.

Mann, M. E., M. A. Cane, S. E. Zebiak, and A. Clement, 2005: Volcanic and Solar Forcing of the Tropical Pacific over the Past 1000 Years. *Journal of Climate*, **18**, 447-456.

McGregor, S., M. Khodri, N. Maher, M. Ohba, F. S. R. Pausata, and S. Stevenson, 2020: The Effect of Strong Volcanic Eruptions on ENSO. *El Niño Southern Oscillation in a Changing Climate*, 267-287.

Ohba, M., H. Shiogama, T. Yokohata, and M. Watanabe, 2013: Impact of Strong Tropical Volcanic Eruptions on ENSO Simulated in a Coupled GCM. *Journal of Climate*, **26**, 5169-5182.

Robock, A., 2020: Comment on “No consistent ENSO response to volcanic forcing over the last millennium”. *Science*, **369**.

Ropelewski, C. F., and M. S. Halpert, 1987: Global and Regional Scale Precipitation Patterns Associated with the El Niño/Southern Oscillation. *Monthly Weather Review*, **115**, 1606-1626.

Stevenson, S., B. Otto-Bliesner, J. Fasullo, and E. Brady, 2016: “El Niño Like” Hydroclimate Responses to Last Millennium Volcanic Eruptions. *J Climate*, **29**, 2907-2921.

Zuo, M., T. Zhou, and W. Man, 2019: Hydroclimate Responses over Global Monsoon Regions Following Volcanic Eruptions at Different Latitudes. *Journal of Climate*, **32**, 4367-4385.

REVIEWERS' COMMENTS

Reviewer #2 (Remarks to the Author):

I would like to thank the authors for their extensive efforts in addressing the comments.

The revised Fig. 2c and Fig. 3d, in particular, have shown a strong evidence of the post-eruption enhancement of the El Niño-EASM relationship, which supports the main message of this work and provides new insights on climatic response to volcanism. Therefore, I recommend the work be accepted for publication.

Reviewer #3 (Remarks to the Author):

This is my second review of this work. The authors have adequately revised the text in response to my and the other reviewers' comments. I therefore recommend accepting the paper, following some minor comments provided below by line number.

98 'quick' leaves too much room for interpretation. Please provide a more quantitative description (e.g., subseasonal).

108 reduced eastward gradient

109 summer, Indo-

144 This statement is speculative. You are not actually showing that the Bjerknes feedback enhances the response. After all, the Bjerknes feedback applies to all Niño responses. You do not show in any way that the Bjerknes feedback has a role in the particular enhancement following eruptions.

186 'current' = modern climate models?

Reviewer #2 (Remarks to the Author):

I would like to thank the authors for their extensive efforts in addressing the comments. The revised Fig. 2c and Fig. 3d, in particular, have shown a strong evidence of the post-eruption enhancement of the El Niño-EASM relationship, which supports the main message of this work and provides new insights on climatic response to volcanism. Therefore, I recommend the work be accepted for publication.

***Response:** We thank the reviewer for raising questions and comments during the first round of review, which helped to clarify the discussion and improve the quality of this manuscript.*

Reviewer #3 (Remarks to the Author):

This is my second review of this work. The authors have adequately revised the text in response to my and the other reviewers' comments. I therefore recommend accepting the paper, following some minor comments provided below by line number.

***Response:** We thank the reviewer for the insightful questions and suggestions during the two rounds of review, which helps to clarify the discussion and improve the quality of this work.*

98 'quick' leaves too much room for interpretation. Please provide a more quantitative [sic] description (e.g., subseasonal).

***Response:** It has been changed to "seasonal"*

108 reduced eastward gradient

***Response:** The description has been corrected as "a **reduced zonal** gradient of Pacific equatorial sea surface temperature (SST)"*

109 summer, Indo-

***Response:** The " ," has been added. [now line 110]*

144 This statement is speculative. You are not actually showing that the Bjerknes feedback enhances the response. After all, the Bjerknes feedback applies to all Niño

responses. You do not show in any way that the Bjerknes feedback has a role in the particular enhancement following eruptions.

***Response:** The decreased SST gradient and weaker easterly winds are shown in supplementary Fig S2. The role of the Bjerknes feedback following volcanic eruptions is discussed in a previous work using the IPSL-CM5B climate model (Khodri et al., 2017). In this work, we did not specifically look into the Bjerknes feedback for all of the model results, and therefore we were refereeing to the existing studies to provide explanation for the observed phenomena in Fig S2.*

Following the reviewer's suggestion, this sentence has been revised to:

*This decreased SST gradient gives rise to a weakened pressure gradient and hence weaker easterly winds and Walker circulation (Supplementary Fig. S2a), which may in turn reduce the SST gradient, a mechanism known as “the Bjerknes feedback”^{33,41}.
(Lines 145-147)*

186 ‘current’ = modern climate models?

***Response:** “the current models” has been revised to “the PMIP3 and CESM models”
(Line 187)*